# Application of a Thermo-Hydrodynamic Model of a Viscous Torsional Vibration Damper to Determining Its Operating Temperature in a Steady State

**DOI:** 10.3390/ma14185234

**Published:** 2021-09-11

**Authors:** Wojciech Homik, Aleksander Mazurkow, Paweł Woś

**Affiliations:** Faculty of Mechanical Engineering and Aeronautics, Rzeszów University of Technology, 35-959 Rzeszów, Poland; whomik@prz.edu.pl (W.H.); almaz@prz.edu.pl (A.M.)

**Keywords:** internal combustion engine, crankshaft, vibration damper, torsional vibrations, temperature, saturation time

## Abstract

The problem of damping torsional vibrations of the crankshaft of a multi-cylinder engine is very important from the point of view of the durability and operational reliability of the drive unit. Over the years, attempts have been made to eliminate these vibrations and the phenomena accompanying them using various methods. One of the methods that effectively increases the durability and reliability of the drive unit is the use of a torsional vibration damper. The torsional vibration damper is designed and selected individually for a given drive system. A well-selected damper reduces the amplitude of the torsional vibrations of the shaft in the entire operating speed range of the engine. This paper proposes a thermo-hydrodynamic model of a viscous torsional vibration damper that enables the determination of the correct operating temperature range of the damper. The input parameters for the model, in particular the angular velocities of the damper elements as well as the geometric and mass dimensions of the damper were determined on a test stand equipped with a six-cylinder diesel engine equipped with a factory torsional vibration damper. The damper surface operating temperatures used in model verification were measured with a laser pyrometer. The presented comparative analysis of the results obtained numerically (theoretically) and the results obtained experimentally allow us to conclude that the proposed damper model gives an appropriate approximation to reality and can be used in the process of selecting a damper for the drive unit.

## 1. Introduction

The main functional assembly in reciprocating piston engines is the crankshaft with connecting rods and pistons. The durability and reliability of the entire engine unit depends on the durability and reliability of the crankshaft–piston system. Due to the friction processes occurring inside, these elements require proper lubrication and constant maintenance of the appropriate thickness of the oil film and conditions of engine operation. This is especially important for crankshaft bearings and the piston and rings assembled with cylinders [1]. Lubrication quality can also be significantly reduced by contaminants entering the lubricating oil from the surrounding air. Therefore, for the long service life of the engine, it is necessary to carefully consider an appropriate design and the efficiency of the intake air filtration [2,3].

Regardless of the design and operational conditions, as well as servicing quality, engine crankshaft–piston assembly still generates adverse vibroacoustic phenomena. These are often treated marginally by the authors of papers on internal combustion engine designs [4,5,6], who focus extensively on aspects such as:Engine designs;Types of ignition;Power supply and cooling system designs;Piston and crank systems;Materials used in the production of engine parts;General methodology of designing selected engine parts, etc.

Meanwhile, the vibroacoustic phenomena accompanying a working engine significantly affect its reliability and durability.

Torsional vibrations of the crankshafts are one of the phenomena accompanying a working engine. The vibrations are caused by the variability of the force *F_T_*—the force tangential to the circle drawn by the crankpin journal (Figure 1). The force causes positive and negative accelerations of the engine crankshaft that change its rotational speed.

The course of the tangential forces *F_T_* (Figure 2) as a function of the angle of rotation of the crankshaft *φ* is most often presented in the form of a tangential force graph [8]. In analyses, the graphs of *F_T_* (*φ*) are often replaced with graphs of unit tangential forces *F_A_* (*φ*).

The unit force *F_A_* (*φ*) is given by:(1)$$F_{A}\left(\varphi \right)\;=\;\frac{F_{T}\left(\varphi \right)}{A}$$
where *A*—cross-sectional area of the cylinder.

The variety of vibration forms and the polyharmonic nature of the tangential force *F_T_* driving the vibrations enable the crankshaft to work in the resonance zone at different rotational speeds of the engine. In a multi-cylinder engine, each family of harmonic exciting forces excited by one cylinder is superimposed on the harmonics of forces excited by the other cylinders. This gives rise to the so-called “enhanced” harmonics, called the main harmonics. For an engine with ignition at equal intervals, the greatest danger is posed by critical rotational speeds at which the order of a harmonic corresponds to the number of ignitions per one revolution of the engine crankshaft, i.e., for two-stroke engines: a multiple of the number of cylinders, and for four-stroke engines: a multiple of half of the number of cylinders [8].

## 2. Torsional Vibrations of the Crankshaft and Methods of Their Elimination

Torsional vibrations of the engine crankshaft are undoubtedly more difficult to detect than others, because they overlap with the rotary motion and often do not cause undesirable vibroacoustic phenomena. Experience proves that the phenomena accompanying torsional vibrations are often incorrectly associated with other shaft vibrations, and consequently, are eliminated with the use of inappropriate methods. We usually learn that the interpretation of these phenomena and the methods used to eliminate them were incorrect when the drive system becomes seriously damaged, e.g., due to a fatigue failure of the shaft (Figure 3).

The phenomena accompanying torsional vibrations of crankshafts drew particular attention for the first time in the shipbuilding industry in the 1920s. Back then, presumably in order to increase the durability of a submarine’s power unit, improve the comfort of the crew′s work, and perhaps above all, to increase the so-called passive defense of the ship (by lowering the acoustic field), a spring-loaded torsional vibration damper, an innovation at that time, was installed on the crankshaft of a submarine’s engine [7]. Despite the long history of marine technology development, the elimination of torsional vibrations of the crankshaft of marine engines still poses a significant problem. Ship engines are multi-cylinder units and thus their crankshafts are relatively long. Therefore, engineers develop increasingly effective methods and algorithms for the detection of torsional vibrations of crankshafts and other power transmission elements in marine propulsion systems with a view to avoiding operating conditions that increase the risk of damage due to failure [9,10,11]. When selecting a torsional vibration damper, efforts are made not only to increase the durability and eliminate torsional vibrations of the crankshaft which are dangerous for the engine design, but also to take into account transient states and reduction in the time delay at step acceleration of the engine rotational speed [12]. The operation of the fuel injection system and the course of the combustion process significantly affect the level of torsional vibrations of the crankshafts in internal combustion engines. Consequently, monitoring of torsional vibrations of the engine shaft may also be an additional effective method for diagnosing the technical condition of the fuel supply system and the proper organization of the combustion process [13,14]. In smaller internal combustion engines, for example, the ones used in passenger cars, the effects of torsional vibrations of the crankshaft are less intense. However, complex drive systems of motor vehicles, such as diesel–electric (hybrid) drives, on the one hand improve the energy efficiency of the source and reduce the emission of harmful exhaust components, but on the other hand, they can generate an increased level of torsional vibrations in the wheel drive system. For this reason, these systems are now attracting increasing interest from researchers in the field of the control and optimization of the torsional vibration level [15].

Over the years, attempts have been made to eliminate torsional vibrations and the accompanying phenomena using various methods, including:Change of rotational speeds of the engine (change of the operating speed range of the engine);Change of the natural frequency of vibrations of the entire system;Change in the course of exciting forces;Use of vibration dampers.

In most cases, the first three solutions were impossible to implement for construction and operational reasons, hence the use of torsional vibration dampers (eliminators), which are most often located at the free end of the engine crankshaft.

Their task is to reduce the amplitude of the resonant torsional vibrations of the engine crankshaft. A well-designed (properly selected, “tuned”) torsional vibration damper allows one to reduce the resonant amplitude of torsional vibrations (by up to 10 times) as well as to shift and reduce the resonance zone (Figure 4).

Over the years, various damper designs have been created, of which viscous vibration dampers have been most widely used, especially in watercraft propulsion systems (Figure 5).

Undoubtedly, this was due to their simple structure, relatively long service life and the fact that, compared to other designs, they work effectively in virtually the entire range of rotational speeds of the engine, enabling it to pass through the so-called critical speeds relatively smoothly (Figure 4) [16].

Torsional vibration dampers are designed or selected individually for a given type of engine.

Before the work on designing a new damper starts, the engine producer should provide the designer with an appropriate set of data, including, among other aspects: information about the engine work cycle (two-stroke, four-stroke engine), the number of cylinders, the cylinder diameter, the piston stroke, the nominal, minimum and maximum rotational speed of the engine, the operating speed range of the engine (if not constant), the ignition sequence, the main journal and crankpin journal diameters, the maximum combustion pressure, the value of the so-called traveling mass (piston and connecting rod masses in reciprocating motion), the inertia and torsional stiffness of the crank throw, the permissible value of the angle of shaft rotation and acceptable dimensions of the damper, and information about the basic units cooperating with the engine, e.g., a generator in the case of power generators. The above data allow, among others, one to:Create a substitute vibrating model of an actual drive system (Figure 6);Determine the course of the tangential forces *F_T_* as a function of the shaft rotation angle *φ* and carry out their harmonic analysis, if the data were not supplied by the engine manufacturer;Determine the basic geometrical parameters of the damper;Calculate the mass moment of inertia of the inertia ring and the damper housing;Determines the size of clearances in the damper;Determine the viscosity of the damping fluid;Calculate the amplitude of resonant vibrations of the free end of the shaft with and without a damper and check the damper thermally in the end phase.

As mentioned before, the purpose of a damper is to effectively reduce the torsional vibrations of the shaft, and thus the vibroacoustic phenomena accompanying them. The results of many years of research and theoretical considerations prove that the relative movement of the inertia ring of a damper in relation to its housing (the damper starts to work) begins when the amplitude of torsional vibrations of the shaft exceeds a certain threshold [16,17]. The start of a relative motion (Figure 7) reduces the amplitude of torsional vibrations of the shaft, while the damper dissipates and converts mechanical energy into heat; as a result, the damper housing begins to heat up.

In the initial period of operation of the damper, the heat generated in it heats up the damper and is partially transferred to the environment. As the temperature of the damper increases, the amount of heat used to heat it up decreases, while the amount of heat transferred to the environment increases. The process continues until a balance is reached between the heat generated and the heat transferred to the environment. The rate at which the damper heats up, called the “damper saturation time”, can therefore be regarded as a measure of the damper effectiveness [17].

The damper saturation time is certainly not a constant value and depends, among other aspects, on the technical condition of the damper, its geometrical parameters, the viscosity of the liquid filling the damper, the rotational speed of the shaft, its vibration level and the environment in which the damper operates.

The above claims are confirmed by the exemplary test results presented in Figure 8, Figure 9 and Figure 10.

As already mentioned above, regardless of the design, a vibration damper is selected individually for a given type of drive unit. Viscous torsional vibration dampers use the phenomenon of friction to dampen vibrations. The mechanical energy dissipated by the damper is converted into heat; therefore, it is recommended to check a dynamically selected damper for the so-called heat loads to avoid overheating during operation. For example, AAM Powertrain (formerly HOLSET, then Metaldyne), a manufacturer of vibration dampers, recommends measuring the heat loads of a damper at engine operation and comparing them with the permissible values. Figure 11 just shows thermal damage of the engine crankshaft vibration damper. Hence, the research on the operating temperature of the damper has a serious practical justification, and the results presented further in the article constitute a significant contribution to the research on viscous torsional vibration dampers and their durability and reliability.

## 3. The Concept of a Thermo-Hydrodynamic Model of a Torsional Vibration Damper

Taking into consideration the thermal phenomena accompanying a working torsional vibration damper, a thermo-hydrodynamic damper model was developed (Figure 12, Formulas (2)–(5)), which enables the determination (calculation) of the housing temperature during its steady-state operation. From the real damper, the model takes information on the main geometric parameters of the damper elements, their kinematics and the occurring force loads. The calculation model includes the equations for the height of the lubricating film, pressure distribution in the lubricating film, and energy balance equations. The thermophysical properties of the oil used were described by the manufacturer′s verified viscosity function. These equations were derived using the principle of conservation of momentum for viscous fluids, the principle of flow continuity, and the principle of energy conservation. When analyzing the operation of the torsional vibration damper, it was assumed that the heat generated by the frictional forces was dissipated from the damper through the housing surface that is directly screwed to the free crankshaft end. Models of a similar type are also used to describe the thermal phenomena in hydrodynamically lubricated slide bearings.

The thermo-hydrodynamic model assumes that:Heat exchange in the damping fluid takes place by convection [18,19,20,21];The value describing the intensity of heat flow through the damper housing to the environment is the heat transfer coefficient α (W/m^2^ °C) (in the first stage of calculations it was assumed that α = const [18]);The model presented in Figure 12 is described by equations taking into account:Geometric parameters (height) of the oil film *h* as a function of relative eccentricity *ε*:
(2)$$h=C_{R}\cdot \left(1+\varepsilon \cdot \cos\varphi^{\prime }\right)\;\;\;where\;\;\;\;\varphi^{\prime }=\varphi -\beta ,\;\;\;\;\;\;\varepsilon =\frac{e}{C_{R}}$$
and where:*C_R_* = 0.5 (*D* − *D_J_*)—radial clearance between housing and ring;*D*—internal diameter of the housing;*D_J_*—outer diameter of the ring;*φ*—angular coordinate;*β*—inclination angle of the center line of the inertia ring and the housing;*e*—ring and housing position eccentricity;*ε*—relative eccentricity.
Pressure distribution in the damper oil film (the equation was derived from the Navier–Stokes equations):
(3)$$\frac{\partial }{\partial z}\left(h{\left(\varphi \right)}^{3}\cdot \frac{\partial p\;\left(\varphi ,\;z\right)}{\partial z}\right)=6\cdot \eta \left(T\right)\cdot \omega_{w}\cdot \;\;\frac{\partial h\;\left(\varphi \right)}{\partial \varphi }\;$$
where:*p*—oil film pressure;*η*(*T*)—dynamic viscosity of the oil;*T*—oil temperature;*ω*_w_ = *ω*_2_ − *ω*_1_—relative angular velocity.

Damper housing temperature T_B_:(4)$$T_{B}=T_{0}+\frac{f_{f}\cdot \omega_{w}\cdot D\cdot F}{\alpha \cdot A_{B}\cdot 2}$$
where: $A_{B}=\pi \cdot D\cdot B_{0}+2\cdot \frac{\pi \left(D^{2}-D_{J}^{2}\right)}{4}$—heat dissipation area;
*T*_0_—T_0_—ambient temperature;*f_f_*—fluid friction coefficient;*F*—ring weight;α—heat transfer coefficient;*B*_0_—width of the housing;

Thermophysical properties of silicone oil M30000 taking into account viscosity changes as a function of temperature according to Clearco Products Co., Inc. (Willow Grove, PA, USA).(5)$$\log\nu \;\left(T\right)=\frac{793.1}{273+T}-2.559+\log\nu \;\left({T|}_{T=25{}^{\circ}C}\right)$$
where *T*—temperature range from 25 °C to 250 °C;


$\nu \left(T\right)\;=\;\frac{\eta \left(T\right)}{\rho \left(T\right)}$—kinematic viscosity;*η*(*T*)—dynamic viscosity according to Figure 13;*ρ*(*T*)—oil density.


## 4. Calculation of the Working Temperature of a Viscous Torsional Vibration Damper

The calculations of the damper housing temperature during steady-state operation were carried out for a damper characterized by the selected geometrical, mass and physical properties presented in Table 1, after performing kinematic analysis (Figure 7). The kinematic analysis of the damper made it possible to determine the value of the relative velocity *ω*_w_ of the inertia ring in relation to the housing, which, as already mentioned before, has a decisive impact on the amount of energy dissipated by the damper.

The calculations of damper housing temperature changes as a function of the relative velocity *ω*_w_ are shown in a graphic form (Figure 14).

## 5. Analysis of Research Results and Conclusions

This paper proposes a thermo-hydrodynamic model of a viscous torsional vibration damper (Figure 12). Despite certain simplifications made in the model, it takes into account the nature of excitation from the drive unit, geometric parameters, as well as physical, kinematic and dynamic properties of the damper. The input parameters for the model, in particular the angular velocities of the damper elements (Figure 7), as well as the geometric and mass dimensions of the damper (Table 1), were determined on a test stand equipped with a six-cylinder diesel engine of the Andoria 6C107 equipped with a factory torsional vibration damper. The damper surface operating temperatures used in model verification (Figure 14) were measured with a laser pyrometer. In order to verify the usefulness of the proposed model in the thermal analysis of the damper, a numerical simulation was performed. The numerical analysis was performed after the data of Table 1 corresponding to the characteristics of a mass-produced damper were entered in the model. The calculation results showed that the proposed thermo-hydrodynamic damper model gives good results of the damper working temperature compared to reality. The range of working temperature of 70–85 °C (according to the inputs and outputs presented in Figure 7 and Figure 14) also proves the damper’s technical state is good and it fits well to the engine. When diagnosing a damper, the theoretical scope of its correct operation can undoubtedly be a valuable source of information. The measurement of temperature of a real object and its comparison with the theoretically determined temperature range allows one to assess the technical condition of a damper, and consequently, its effectiveness in vibration damping.

## Figures and Tables

**Figure 1 materials-14-05234-f001:**
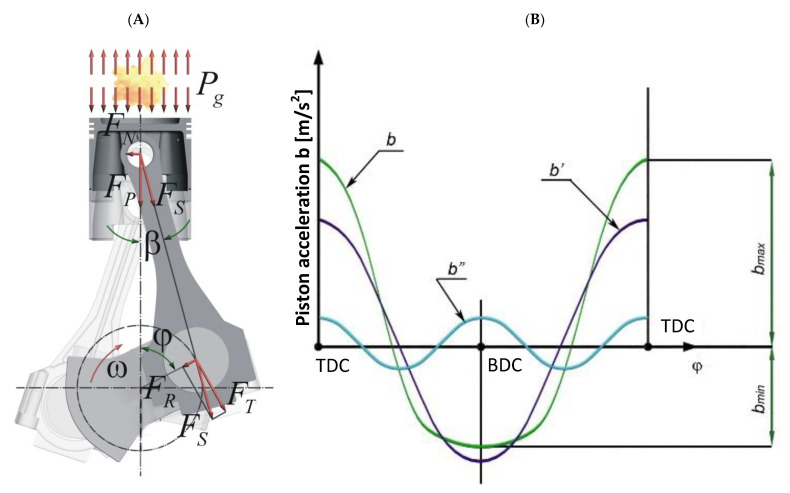
Crankshaft—piston assembly layout (**A**) and changes in acceleration of piston (**B**) as a function of crankshaft position angle *φ* [7]; b—total acceleration rate of the piston; b’, b”—first and second order accelerations; *F_N_*—normal force, *F_T_*—force tangential to the circle drawn by the crank throw, *F_R_*—radial force acting along the momentary position of the crank arm, *F_S_*—resultant force, *P_g_*—combustion gas pressure, *β*—inclination angle of the resultant force, *φ*—crankshaft position angle, *ω*—crankshaft angular velocity, TDC, BDC—top and bottom return positions of the piston.

**Figure 2 materials-14-05234-f002:**
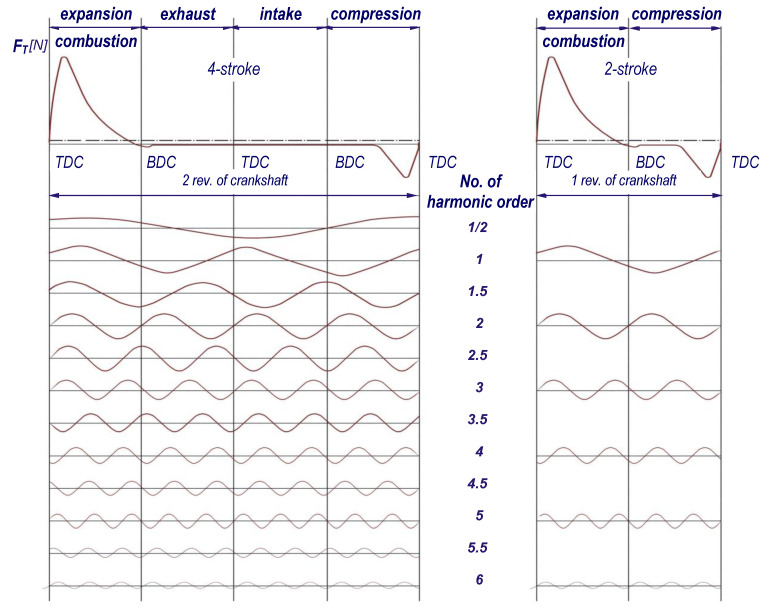
The course of the tangential force *F_T_* and its successive harmonics in a four- and two-stroke engine [7,8].

**Figure 3 materials-14-05234-f003:**
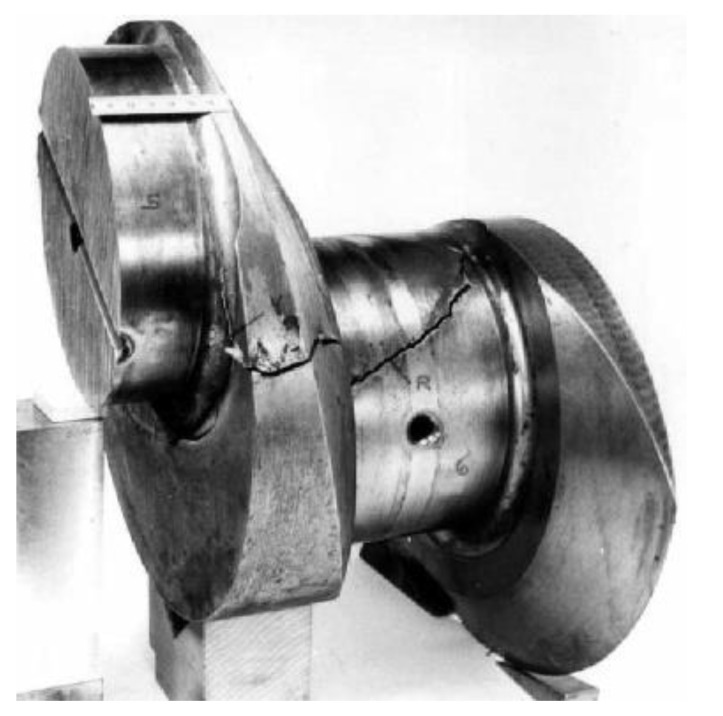
The crankshaft journal damaged by torsional vibration loads [7].

**Figure 4 materials-14-05234-f004:**
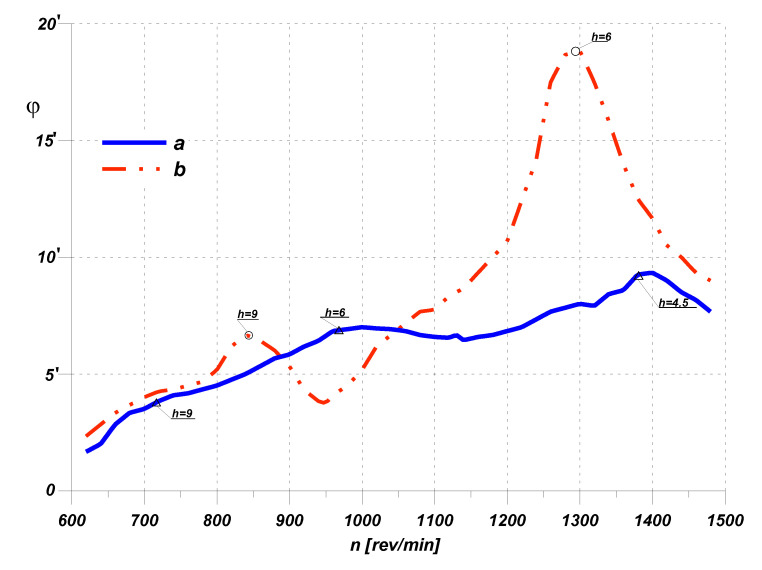
Torsional vibrations of the shaft—damper effectiveness: a—with a damper, b—without a damper [7,8]; *h*—number of harmonic order of vibrations.

**Figure 5 materials-14-05234-f005:**
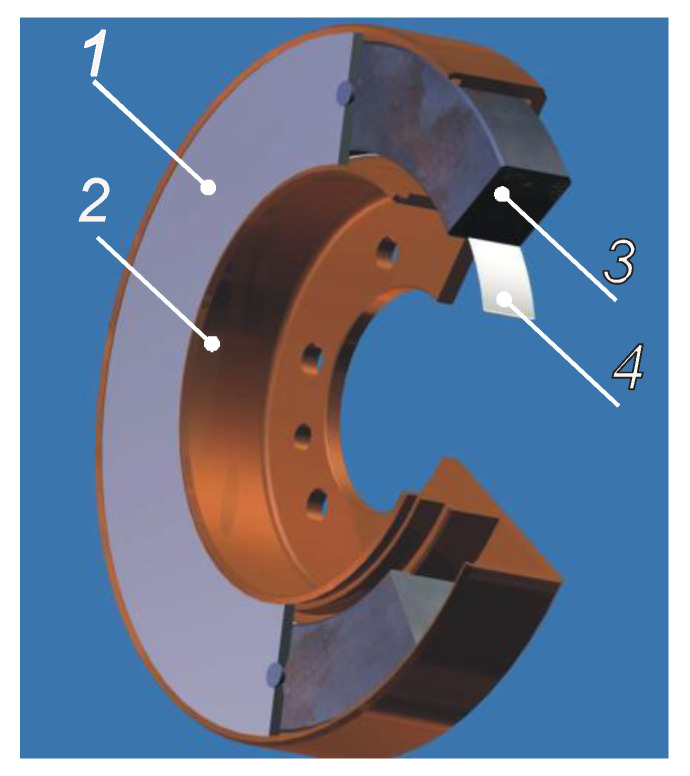
Viscous vibration damper: 1—cover, 2—housing, 3—inertia ring, 4—bearing [7].

**Figure 6 materials-14-05234-f006:**
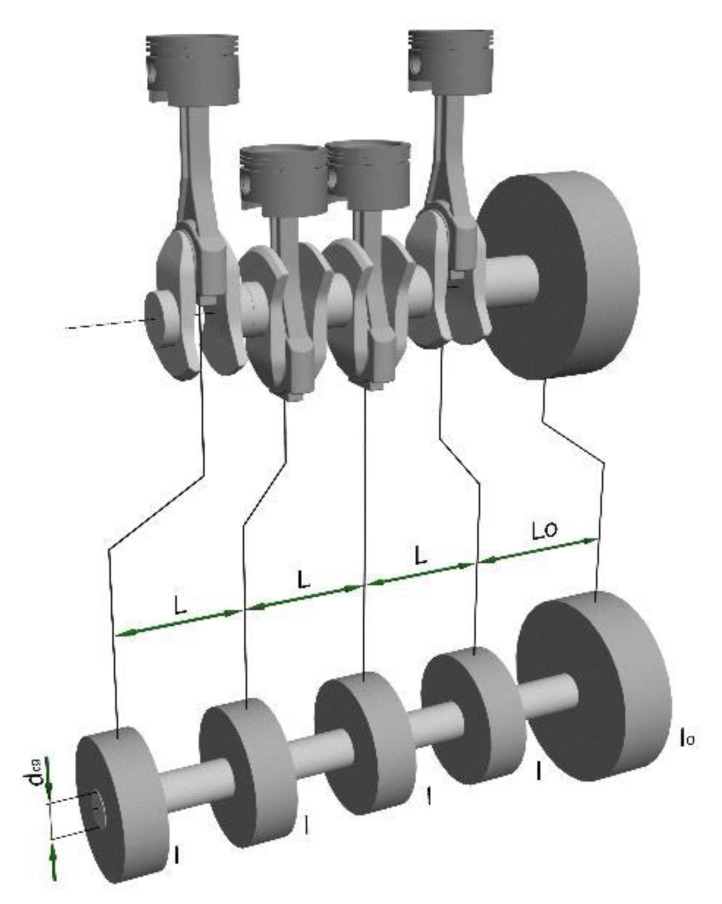
The actual and substitute model of a crank–piston system [8]; I—reduced mass of inertia of individual crank–piston systems, I_0_—reduced mass of inertia of the flywheel, L—reduced distances between the reduced masses of inertia for the crank–piston systems, L_0_—reduced distance between the reduced mass of inertia of flywheel and the last crank–piston system, d_cp_—reduced diameter of the crankshaft.

**Figure 7 materials-14-05234-f007:**
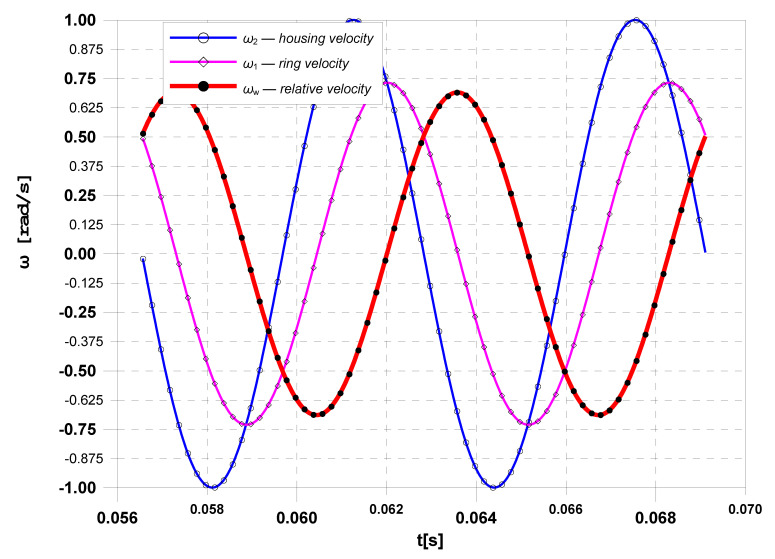
Angular velocity charts: *ω*_2_—of the housing, *ω*_1_—of the ring, *ω*_w_—of relative angular velocity [8].

**Figure 8 materials-14-05234-f008:**
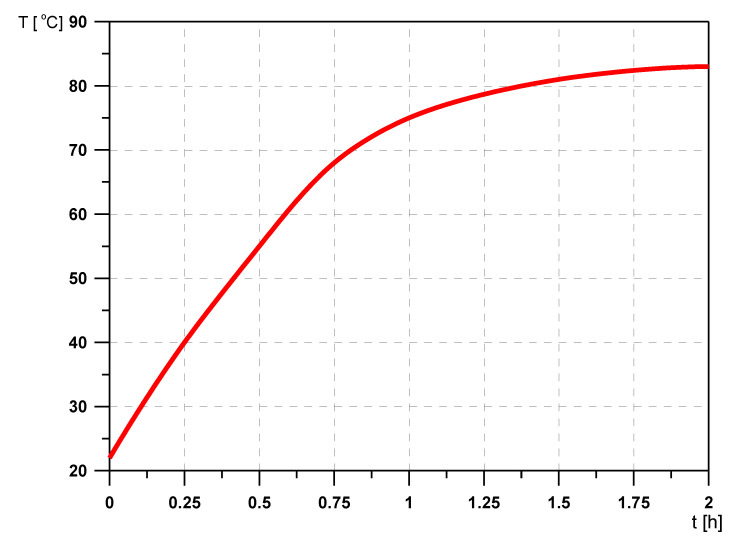
Damper temperature changes as a function of time: filled with a liquid with a viscosity of 30,000 cSt.

**Figure 9 materials-14-05234-f009:**
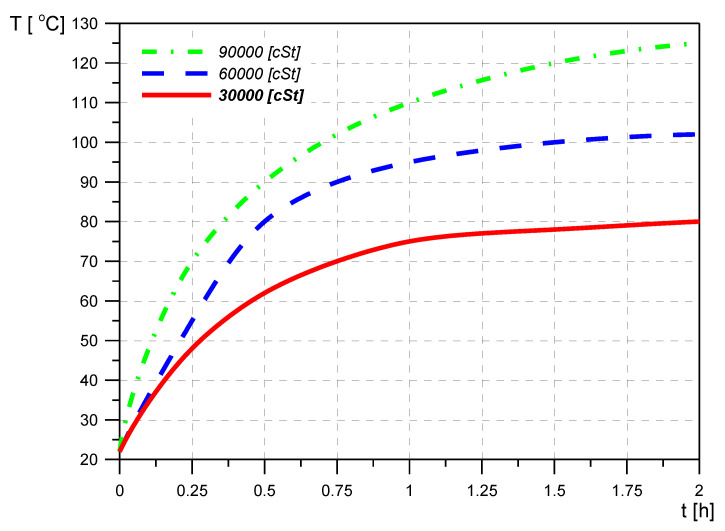
Temperature changes of a damper filled with liquids of different viscosities as a function of time.

**Figure 10 materials-14-05234-f010:**
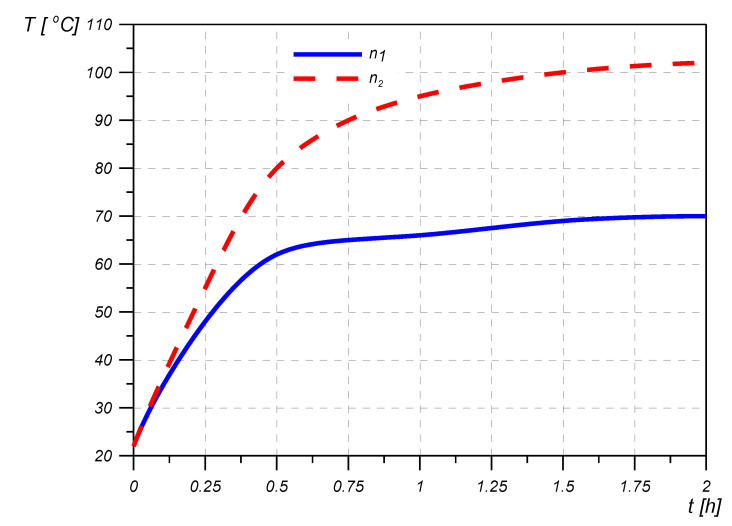
Damper temperature changes as a function of time for different shaft rotational speeds n_i_ (n_1_ < n_2_).

**Figure 11 materials-14-05234-f011:**
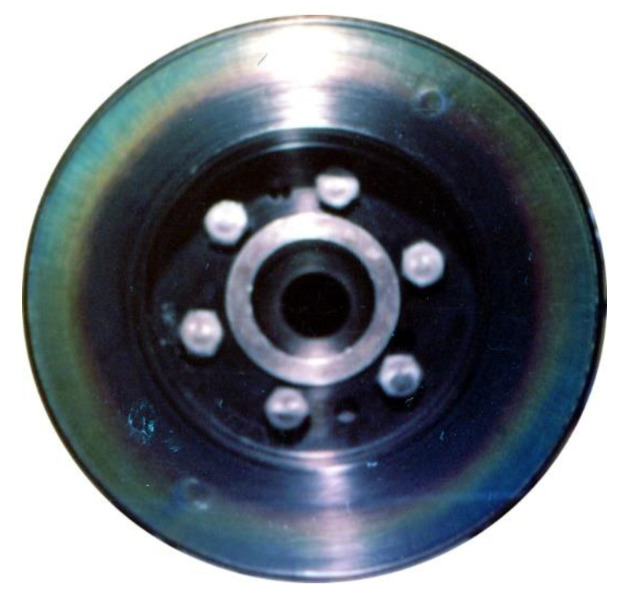
Overheated viscous torsional vibration damper [7].

**Figure 12 materials-14-05234-f012:**
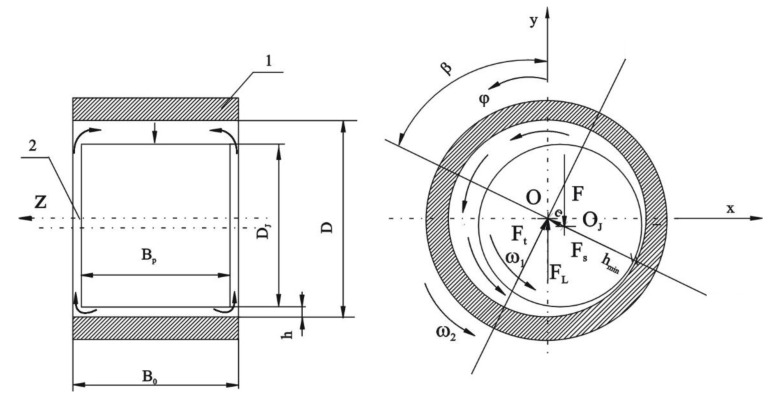
The model including basic geometry, kinematics, oil flow directions and forces in the torsional vibration damper; 1—housing, 2—inertia ring, *D*—internal diameter of the housing, *D_J_*—outer diameter of the ring, *B_0_*—width of the housing, *B_p_*—width of the ring, *h*—oil film thickness, *x,y,z*—axes of Cartesian coordinate system, *φ*—angular coordinate, *ω*_1_—angular velocity of the ring, *ω*_2_—angular velocity of the housing, *O*—center of the housing, *O_J_*—center of the ring, *F*—ring weight, *F_L_*—the force of hydrodynamic buoyancy with its component forces—normal (*F_t_*) and tangential (*F_s_*), *h_min_*—minimal oil film thickness, *e*—ring and housing position eccentricity, *β*—inclination angle of the center line of the inertia ring (*O_J_*) and the housing (*O*).

**Figure 13 materials-14-05234-f013:**
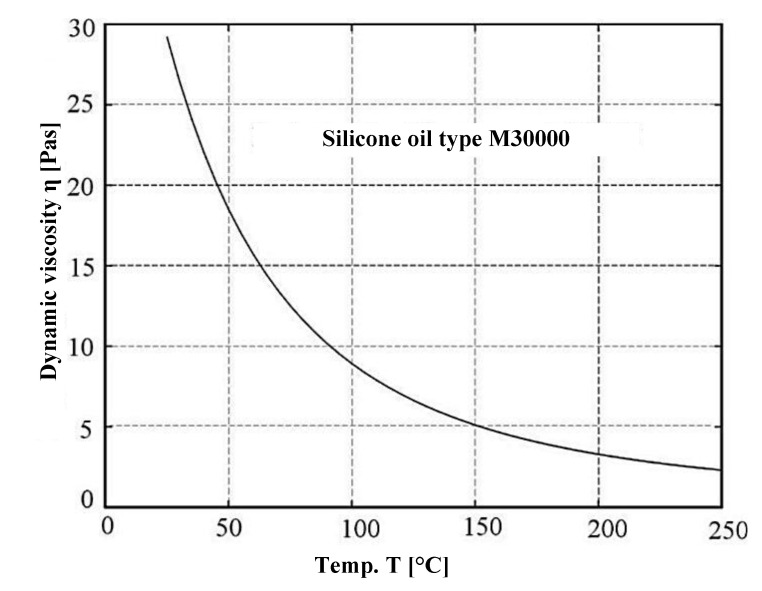
Dynamic viscosity of oil M30000 as a function of temperature.

**Figure 14 materials-14-05234-f014:**
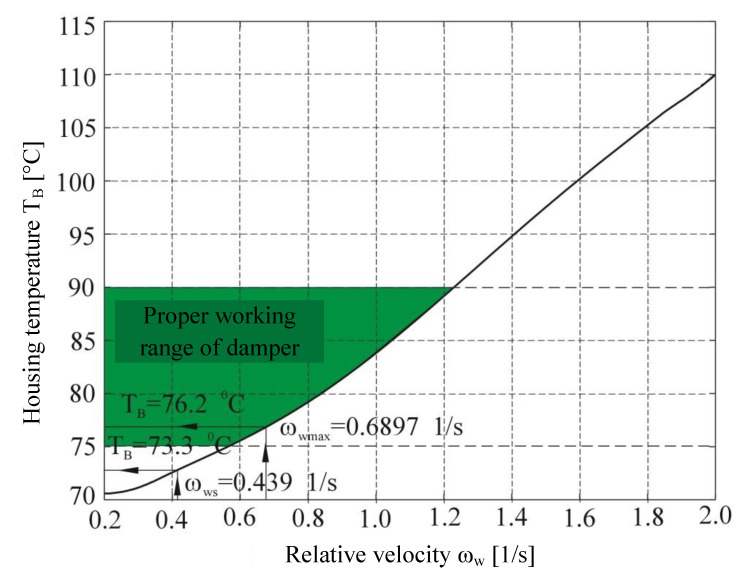
Calculated rate and experimental data for working temperature of a viscous torsional vibration damper T_B_ as a function of the relative velocity of the inertia ring; *ω*_ws_—measured average relative velocity of the inertia ring, *ω*_wmax_—measured maximum relative velocity of the inertia ring.

**Table 1 materials-14-05234-t001:** Main damper parameters taken for calculations.

Geometric, Physical and Kinematic Parameters.	
1. Outer diameter of the ring	*D_J_* = 2*R_J_* = 207.925 mm
2. Internal diameter of the housing	*D* = 2*R* = 208.109 mm
3. Radial clearance	*C_R_* = *R* − *R*_j_ = 0.092 mm
4. Inertia ring width	*B_p_* = 33.00 mm
5. Inertia ring weight	*F* = 89.6 N
6. Relative angular velocity	*ω_w_* = *ω*_2_ − *ω*_1_ = 0.4–2.0 1/s
7. Kinematic viscosity of silicone oil	*ν* = 30,000 cSt
8. Oil density	*ρ* = 970 kg/m^3^

## Data Availability

The data presented in this study are available on request from the corresponding author.

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
