# Peer review of "Application of a Thermo-Hydrodynamic Model of a Viscous Torsional Vibration Damper to Determining Its Operating Temperature in a Steady State"

_materials, 2021, doi:10.3390/ma14185234_

Round 1

Reviewer 1 Report

The paper is not well-written, and I do not think it is publishable in its current form. Some concerns are as follows:

Does Section 3 represent the new model proposed in this paper? Where do the equations come from? How are they derived? What does each variable denote?

What is the novelty of the proposed model compared to existing models? Are the authors sure the model is new and has not been used by others before?

How do the authors verify the correctness of the model? The authors claim that “the presented comparative analysis of the results obtained numerically (theoretically) and the results obtained experimentally …”. However, it is unclear where the comparisons are made in the manuscript.

10 references out of the 18 cited references are not published in English and are not accessible to general readers.

Reviewer 2 Report

Uwagi w załączniku

Reviewer 3 Report

The paper discusses an important subject. The advances in the data-acquiring process and the large extend of systems necessitates having efficient tools, such as the one proposed in this draft. The paper is well written and organized. My comments are as follows:

- While the abstract has aimed to provide a comprehensive overview of the main contribution, there is a need to be revised so that the general reader can grasp the main idea/topic of the draft and the main contribution. 

- Having a nice schematic diagram in the draft would be helpful. This alleviates the difficulty of going to details of the techniques for the readers.

- There has been a surge in the application of Machine Learning and Statistical framework to solve similar problems focused in this paper. The authors are encouraged to include some of the recent articles in the introduction to give an excellent holistic overview of the existing techniques to general readers:

# "Recent trends on nanofluid heat transfer machine learning research applied to renewable energy." Renewable and Sustainable Energy Reviews (2020): 110494.

# "Partially-Observed Discrete Dynamical Systems", 2021 American Control Conference (ACC), 2021.

# "On the tilting‐pad thrust bearings hydrodynamic lubrication under combined numerical and machine learning techniques." Lubrication Science 33.3 (2021): 153-170.

- The format of some of the references is not in standard form. These need to be checked and fixed.

Round 2

Reviewer 1 Report

To be published in a peer-reviewed journal, the proposed method must be verified and validated in a very rigorous manner. In this regard, The authors' slight revisions do not make much improvement. Therefore, the paper should be rejected.

Reviewer 2 Report

The improvement was doThe improvement was done in line with my expectationsne in line with my expectations

Reviewer 3 Report

The paper is well-revised and in my opinion, it is ready for publication.